# Barriers and facilitators to facility-based delivery in rural Zambia: a qualitative study of women's perceptions after implementation of an improved maternity waiting homes intervention

Rachel M Fong [1] , Jeanette L Kaiser [1] , Thandiwe Ngoma [2] , Taryn Vian [3], Misheck Bwalya [4] , Viviane Rutagwera Sakanga [5] , Jody R Lori [6], Kayla J Kuhfeldt [1] , Gertrude Musonda [7] , Michelle Munro-Kramer [8], Peter C Rockers [1] , Davidson H Hamer [1,9] , Eden Ahmed Mdluli [10], Godfrey Biemba [11] , Nancy A Scott [1]

For numbered affiliations see end of article.

**Correspondence to**
Ms Rachel M Fong;
rmfong@bu.edu

## ABSTRACT

**Objectives** Women in sub-Saharan Africa face well-documented barriers to facility-based deliveries. An improved maternity waiting homes (MWH) model was implemented in rural Zambia to bring pregnant women closer to facilities for delivery. We qualitatively assessed whether MWHs changed perceived barriers to facility delivery among remote-living women.

**Design** We administered in-depth interviews (IDIs) to a randomly selected subsample of women in intervention (n=78) and control (n=80) groups who participated in the primary quasi-experimental evaluation of an improved MWH model. The IDIs explored perceptions and preferences of delivery location. We conducted content analysis to understand perceived barriers and facilitators to facility delivery.

**Setting and participants** Participants lived in villages 10+ km from the health facility and had delivered a baby in the previous 12 months.

**Intervention** The improved MWH model was implemented at 20 rural health facilities.

**Results** Over 96% of participants in the intervention arm and 90% in the control arm delivered their last baby at a health facility. Key barriers to facility delivery were distance and transportation, and costs associated with delivery. Facilitators included no user fees, penalties for home delivery, desire for safe delivery and availability of MWHs. Most themes were similar between study arms. Both discussed the role MWHs have in improving access to facility-based delivery. Intervention arm participants expressed that the improved MWH model encourages use and helps overcome the distance barrier. Control arm participants either expressed a desire for an improved MWH model or did not consider it in their decision making.

**Conclusions** Even in areas with high facility-based delivery rates in rural Zambia, barriers to access persist. MWHs may be useful to address the distance challenge, but no single intervention is likely to address all barriers experienced by rural, low-resourced populations. MWHs should be considered in a broader systems approach to improving access in remote areas.

## STRENGTHS AND LIMITATIONS OF THIS STUDY

⇒ The study was conducted after the implementation of a large-scale maternity waiting home intervention, which allows us to examine the differences in women's perceptions of barriers experienced between areas that received the Core Maternity Waiting Home Model and areas that did not receive the intervention.

⇒ We used a randomly selected, representative sample of the population, which is rare in qualitative research, and it assured the best opportunity to capture a diversity of perceptions.

⇒ Our sample of women living in rural Zambia is a hard-to-reach population that likely face the greatest barriers to facility delivery due to their low socioeconomic status, poor road infrastructure, limited transportation options, and low densities of facilities and skilled birth attendants.

⇒ The findings are subject to recall bias, as we interviewed women who delivered up to 12 months prior to data collection; however, topics of discussion surrounded social norm, which is less subject to recall bias than personal experiences and decision making.

⇒ We conducted multiple rounds of data collection throughout the duration of the study, which likely resulted in increased awareness of maternity waiting homes among the control groups.

**Trial registration number** NCT02620436.

## INTRODUCTION

To reduce maternal and neonatal mortality, the World Health Organization (WHO) recommends that every birth occurs at an equipped health facility, attended by a skilled professional.[1] However, home births remain

common in many areas, particularly in rural, low-resource settings. Individual and system level factors associated with births outside of equipped health facilities are well documented and include lack of access to facilities, long distances, lack of transportation, costs associated with delivery, and women's decision-making autonomy.[2–7]

The Government of the Republic of Zambia (GRZ) has sought to increase facility-based deliveries and reduce maternal mortality through an increasingly articulated policy landscape, monitoring of relevant indicators and strategic partnerships to impact supply and demand of maternal health services. Within its National Health Strategic Plan (NHSP), which is published approximately every 5 years and sets out the country's health goals over that period, the GRZ has emphasised the importance of deliveries supervised by skilled health personnel. This has been included as a monitoring indicator since at least 2005, and subsequently, as a main objective of the NHSP since at least 2010.[8 9] Additionally, in 2006, the GRZ eliminated user fees on primary health services, including maternal and child health services, first in rural areas then countrywide, to remove cost barriers and increase universal access to health services.[10]

The GRZ more clearly articulated its commitment to reducing maternal mortality and increasing maternal health more generally through its 'Roadmap for accelerating the reduction of maternal, newborn and child mortality,' which mapped all stakeholder groups and organisations within the country and set out a multi-pronged, ambitious plan for achieving its goals.[11] Through the Saving Mothers, Giving Life (SMGL) initiative, formed in 2011, which combined GRZ efforts with multiple external funders and implementers, many of the plans set out in the roadmap were implemented. Between 2012 and 2016, SMGL significantly increased facility-based delivery rates from 63% to 90% and reduced maternal mortality by approximately 40% in multiple districts of Zambia through intensive investments in increasing access to and improving quality of health services, care coordination and community outreach programmes.[12 13] These same results were not experienced in non-SMGL-supported districts.[12]

One strategy to increase facility-based deliveries is maternity waiting homes (MWHs), which in theory allow a pregnant woman to reside near a health facility in the weeks prior to her estimated delivery date.[7 14–16] With lower densities of equipped health facilities in rural areas, and poor road infrastructure, women residing in rural locations typically endure lengthy travel to deliver at a well-equipped health facility and often experience challenges accessing transportation.[17–19] In rural settings, the presence of quality MWHs has been associated with increased rates of facility-based delivery.[14 20–22]

Through formative research conducted in Zambia, the Maternity Homes Alliance (MHA), a partnership between the GRZ, two local non-governmental organisations (Right to Care Zambia and Africare), and two academic institutions (Boston University and the University of Michigan), designed a Core MWH Model targeting pregnant women living most remotely.[23 24] The MHA hypothesised that quality MWHs could remove distance barriers to facility delivery by allowing women to travel to facilities earlier than when they are in labour or when it becomes urgent. The MHA constructed 20 MWHs following the Core MWH Model (referred to hereafter as the Core MWH Model) and evaluated the impact on facility delivery rates among women living more than 10 km from their designated rural health facility.[25] The Core MWH Model was associated with increased rates of facility-based delivery and MWH utilisation among remote-living women.[22 26] The MHA also evaluated the implementation effectiveness of the Core MWH Model to assess implementation fidelity and factors that affect adoption, uptake and sustainability of the intervention.[27] A mixed-methods substudy looking at the quality and utilisation of MWHs at referral facilities found that the Core MWH Model had a significantly higher-quality score than comparison site MWHs, and there was an increase in MWH utilisation at both intervention and comparison sites, though there was a higher increase at one of the two intervention sites, after the implementation of the intervention.[28] As part of the larger impact evaluation, this study aimed to understand how the Core MWH Model influenced maternal perceptions of barriers and facilitators to facility-based delivery by conducting a qualitative assessment among women living in villages 10+ km from the health facility in catchment areas that received the Core MWH Model compared with those living in control catchment areas without the improved MWHs.

## METHODS
### Study setting
This study was conducted in seven rural districts in Zambia which received the SMGL intervention: Nyimba and Lundazi in Eastern Province; Choma, Kalomo and Pemba in Southern Province; and Mansa and Chembe in Luapula Province.[25] These districts are generally rural, with low population densities, and their main industries are agriculture.[29] The majority of residents, especially away from the few urban centres, are subsistence farmers who may make some additional income selling crops or doing piecework (short-term, casual work).[30]

The mixed-methods baseline evaluation for this study found high facility delivery rates of 81.4% among women living more than 10 km from their rural health facility, with 15.3% of women still delivering at home and 3.2% delivering on the way to a health facility.[19] Qualitatively, these remote-living women discussed transport availability, distance, and cost of transport and supplies as continued barriers to facility-based deliveries. They also noted informal penalty fees being levied by traditional leaders against women who delivered at home, which had been previously reported in the literature.[19 31]

## Intervention description

Described fully elsewhere, the MHA constructed 20 MWHs that met each of the three main pillars of the Core MWH Model:

1. Infrastructure, equipment and supplies to ensure a safe, comfortable and functional structure. The Core MWH Model is a multiroom structure with concrete floors and walls, a secure roof, and lockable doors and windows. The structure includes residential space for pregnant women, a separate pace for postnatal women and newborns, latrines, a bathing area, a cooking space and a veranda for socialising and education sessions. Amenities include lighting, cooking supplies, beds, bedding, mosquito nets and storage space.
2. Policies, management and finances to ensure local oversight and sustainability of the homes. Each MWH has a management unit and a governance committee which includes members of the community, health facility staff and local government. These oversight organisations oversee daily operations and maintain financial well-being and sustainability of the MWH.
3. Linkages and services to ensure integration with the formal health system. All clinical care was provided at the health facility, but health facility staff jointly oversaw the MWHs as part of its governing committee and checked in daily on waiting women. Education courses on maternal and child health topics were conducted weekly at each intervention site.[25]

The implementation of the Core MWH Model in all 20 sites was similar, with minor differences in how the partners and individual sites operationalised the policies, management and finances pillar. For example, income-generating activities—such as maize grinding, goat raising or a store—were implemented at intervention sites to help finance and sustain the MWHs; each site implemented its own income-generating activity.

Twenty rural health facilities implementing the standard of care for waiting women were selected as matched control sites.[25] The standard of care varied widely with some having no formal space for waiting women and others having a small, community-constructed MWH in generally poor condition and minimal amenities, if any.

## Study design

Data for this analysis were collected from September to October 2018 as part of the endline data collection of the quasi-experimental, cluster-controlled impact evaluation.[25 32 33] A household survey was administered to a randomly selected cross-section of 2330 participants, 1217 in the intervention, 1113 in the control sites who lived 10 or more km from their designated rural health facility. A randomly selected subsample of participants (7%) also completed an in-depth interview (IDI) following the survey. The semistructured IDI guide was based off of the household survey questions (online supplemental file 1). It had a series of open-ended questions with prompts to explore themes on personal and community perceptions of factors that influence delivery location, quality of MWHs, factors that influence utilisation of MWHs, planning for delivery and costs related to delivery. The questions included probes for various themes that arose during baseline data collection and known facilitators and barriers from existing literature, including culture, cost, distance, transportation, safety, comfort and local laws, among others.[19] The intervention MWHs were launched shortly after baseline data collection (March to May 2016) of the impact evaluation and operated for a minimum of 13 months before endline data collection.

## Participants

A random sample of participants were asked to participate in the IDI following their household survey. Per the primary study eligibility criteria, all participants were at least 15 years of age or older and delivered a baby in the past 12 months, irrespective of location of delivery. To mitigate selection bias, we first randomly selected four villages per catchment area, then randomly selected one household in each village to participate in the IDI following the main survey. This helped to select a representative sample of women in the sample frame.[25]

## Data collection

Data collectors fluent in English and the four relevant local languages were trained in human subjects' protection, electronic data collection using Android tablets and qualitative research methods. Immediately following the quantitative household survey, selected participants were offered a short break before commencing the IDI, which took approximately 30 min to complete. All interviews were audiorecorded with consent from the participant. Participant demographics, collected during the household survey, were linked to IDI participants through their unique study ID. Demographics were captured electronically on encrypted tablets using the SurveyCTO Collect software (Dobility, Cambridge, Massachusetts, USA).[34]

## Data management and analysis

All IDI recordings were translated and transcribed verbatim into Microsoft Word. The transcripts were coded line by line in NVivo V.12 (QSR International, Doncaster, Australia) by two research staff familiar with the context and study.[33 35 36] The main codes were created a priori using the established themes from the interview guide. Additional subcodes were added during coding as themes emerged. The first few transcripts were coded by both research staff and a coding comparison was run in NVivo V.12 (QSR International, Doncaster, Australia) using a kappa coefficient to mitigate researcher bias. Codes with poor agreement (kappa value <0.40) were discussed and agreed on. Codes were reviewed throughout the coding process and merged if there was agreement that two or more codes contained similar content. Discrepancies were deliberated and resolved. Matrix queries in NVivo V.12 (QSR International, Doncaster, Australia) were used during the analysis to organise and understand themes in the coded data to compare perceived facilitators and

barriers to facility delivery between remote, rural women living in catchment areas with the Core MWH Model and those in areas without the Core MWH Model. Themes with more than 10 participants (25% of the sample) were summarised and included in the results. Findings were triangulated within the context of the bigger impact evaluation. We interpreted findings within the Health Belief Model, which explains health behaviours as the presence or absence of multiple constructs, including perceived benefits and barriers to a behaviour.[37] Quotations which included costs in Zambian Kwacha (K) were converted to US Dollars (US$) using an approximate average exchange rate over the 12 months preceding data collection (K10 to US$1). Findings were reported using the Standards for Reporting Qualitative Research Checklist.[38]

Demographic data were cleaned and analysed in SAS V.9.4 (SAS Institute).[32] We present demographics for the recently delivered women and their households and the women's primary outcomes of delivery at a health facility. We report the median and IQR for non-normal data, means and SD or proportions where appropriate.

## Patient and public involvement

Key community members including women, men, elders and traditional leadership were consulted in the conceptualisation and design of the Core MWH Model.[23 24] The final design also included input from stakeholders at all levels throughout the Ministry of Health. Through a process evaluation during implementation, end-users and the public were regularly consulted to understand the acceptability and sustainability of the intervention.[39–42]

## RESULTS
### Participant characteristics

Demographic characteristics for the 158 IDI participants were similar between the intervention and control groups (table 1). Participants in both arms lived a similar distance from their assigned health facility (12–13 km) and had about 6 years of education. More intervention participants (65.4%) reported having used an MWH than in the control arm (42.5%). More participants in the intervention arm delivered at a health facility (96.1%) compared with the control arm (90.0%).

### Factors influencing delivery location

Two major barriers influencing facility-based delivery emerged: (1) distance and transportation, and (2) costs associated with delivery, while four key facilitators emerged: (1) no user fees, (2) penalties for home delivery, (3) desire for safe delivery and (4) availability of MWHs (table 2).

### Distance and transportation

Participants in both study arms regarded distance and transportation as key barriers when deciding whether or not to deliver at a health facility. Participants generally acknowledged the health facility is far, so they have

**Table 1** Qualitative participant characteristics for the endline observation of the MWH impact study, by study arm

|  | Intervention n=78 | Control n=80 |
|---|---|---|
| Participant characteristics | | |
| Woman's age in years, median (IQR) | 26 (21–32) | 29.5 (23–35) |
| Years of education, mean (SD) | 6.1 (3.2) | 6.0 (3.4) |
| Married/cohabiting, n (%) | 68 (87.2) | 71 (88.7) |
| Gravida, mean (SD) | 4.0 (2.4) | 4.5 (2.3) |
| Parity, mean (SD) | 3.5 (2.4) | 4.2 (2.3) |
| Primigravida, n (%) | 16 (20.5) | 10 (12.5) |
| Four or more ANC visits, n (%) | 60 (75.0) | 59 (75.6) |
| Delivered at health facility or hospital, n (%) | 75 (96.1) | 72 (90.0) |
| Age of most recently delivered baby (months), mean (SD) | 7.0 (3.7) | 7.1 (3.6) |
| District, n (%) | | |
| Choma/Pemba | 15 (19.2) | 11 (13.7) |
| Kalomo | 16 (20.5) | 21 (26.2) |
| Nyimba | 8 (10.3) | 8 (10.0) |
| Lundazi | 20 (25.6) | 20 (25.0) |
| Mansa/Chembe | 19 (24.4) | 20 (25.0) |
| Household characteristics | | |
| Household size, median (IQR) | 6 (5–8) | 7 (5–8) |
| Dependency ratio*, mean (SD) | 1.5 (1, 2) | 1.5 (1, 2) |
| Travel distance to health facility (km), median (IQR) | 12.1 (11–15) | 13.2 (11–16) |
| No electricity, n (%) | 77 (98.7) | 79 (99.0) |
| Unimproved sanitation†, n (%) | 66 (84.6) | 59 (73.7) |

*Dependency ratio = (children <16 years old +adults >65 years old)/adults >16 years old.
†Unimproved sanitation: pit latrine without slab/open pit, bucket toilet, hanging toilet/latrine, no facility/bush/field.
ANC, antenatal care; MWH, maternity waiting home.

to prepare transportation (eg, booking a car or taxi, borrowing a bicycle or ox cart) and put aside money in advance (table 3, quotes a-d). No facilitators relating to distance and transportation arose in either the intervention or control groups.

### Costs associated with delivery

Delivery-associated costs were a frequently cited barrier. Nuances to these costs, such as costs associated with having to bring food for their MWH stay before delivery and stigma related to not having enough money to purchase delivery supplies were also captured. Many participants reported that procuring delivery supplies posed an obstacle to a facility-based delivery (table 3, quotes e and f). A few participants mentioned that one reason they still deliver at home is due to fear of getting embarrassed or shouted at by the health facility staff if they are unable

**Table 2** Key barriers and facilitators to facility-based delivery reported by qualitative participants during the endline observation of the MWH impact study, by study arm

| Key themes | Intervention | Control |
|---|---|---|
| Barriers influencing facility delivery | 1. Distance and transportation: health facility is far, have to prepare transport and money, buy fuel in advance or walk a long way; bad roads<br>2. Costs associated with delivery: need to prepare money for transportation (costs more if distance is farther) and buy required delivery supplies<br> a. Costs associated with MWH stay: need to bring food to cook during stay<br> b. Stigma related to costs: worried, scared or embarrassed about not having the delivery supplies required and/or baby clothes to bring to the health facility | 1. Distance and transportation: health facility is far and transportation is challenging, have to plan to walk early and wait at the facility or MWH if there is one or need to prepare transport and money<br>2. Costs associated with delivery: need to prepare transport and money for transportation and buy required delivery supplies<br> a. Stigma related to costs: scared or embarrassed about not having delivery requirements and/or baby clothes to bring to the health facility |
| Facilitators influencing facility delivery | 1. Costs associated with delivery: no cost to deliver at a facility; no cost to stay at MWH<br>2. Penalties for home delivery: penalty fee for home delivery—pay the chief and health facility; penalty made women fear home delivery though some believed it is a hoax<br>3. Safe delivery: concerned about complications at home and afraid of dying; safer to deliver at facility, where staff can help with complications<br>4. MWH availability: good to move to MWH before delivery to avoid transportation difficulties because health facility is far<br> a. Advised to stay by health facility staff at 8 months if woman lived far away<br> b. MWH is free and has amenities such as beds, blankets and cooking supplies | 1. Costs associated with delivery: no cost to deliver at a facility<br>2. Penalties for home delivery: penalty fees for home delivery—pay the chief and health facility; penalty made women fear home delivery<br>3. Safe delivery: concerned about delivery complications at home, and afraid of dying from complications; safer to deliver at facility, where staff can help with complications<br>4. MWH availability: women chose to go to health facilities with MWH no matter how far; good for those who live far away from the facility—can rest before delivery; no MWH but want one in their area; MWH availability does not matter, women will go where they can deliver safely |

MHW, maternity waiting homes.

**Table 3** Quotations illustrating key themes as barriers or facilitators to facility-based delivery as reported by qualitative participants during the endline observation of the MWH impact study, by study arm

| Key themes | Theme as barrier versus facilitator to facility delivery | Intervention | Control |
|---|---|---|---|
| Distance and transportation | Barrier | (a) "Yes, that's the reason women go to the MWH early, transport is a big major challenge." – Woman, Pemba District (b) "[The health facility] is quite far. We walked there. I started off around 3pm and reached there at 7pm." – Woman, Lundazi District | (c) "The distance from here to the clinic is quite long and transport here is a big challenge, because you need to have money to pay." – Woman, Pemba District (d) "The distance [to the health facility] is very big and transport is a problem…" – Woman, Lundazi District |
| | Facilitator | No themes emerged | No themes emerged |
| Costs associated with delivery | Barrier | (e) "I thought about where I was going to find money to get what is required. They tell you exactly what you need to buy when going for antenatal checkups so that you can prepare." – Woman, Lundazi District | (f) "The basic costs that we face are buying baby clothes, delivery kit, transport, and buying food when you stay at the clinic as you wait to deliver." – Woman, Lundazi District (g) "This is why people deliver at home, because most have no money to prepare for delivery at facilities and when you get there without these (required items), nurses shout at us a lot." – Woman, Nyimba District |
| | Facilitator | (h) "[The health facility] does not charge. They only ask us to prepare what to use during delivery and for our children too." – Woman, Chembe District | (i) "That's why we prefer delivering from a health facility where it is free than in the village where you have to pay the village headman and other people." – Woman, Chembe District |
| Penalties for home delivery | Barrier | No themes emerged | No themes emerged |
| | Facilitator | (j) "If you deliver at home, like me, when you go to the health facility to have the baby checked you will need to pay K200 (~US$20) before your baby is checked. The nurses are the ones who give these penalties." – Woman, Kalomo District | (k) "We have our traditional headmen who have partnered with the health facilities. If you deliver at home, you will pay a goat to the chief and then when you go to the health facility, they will make you pay K100 (~US$10) to get the growth monitoring card (for the baby). Those are the local rules that we do follow." – Woman, Lundazi District |
| Safe delivery | Barrier | No themes emerged | No themes emerged |
| | Facilitator | (l) "Different things(like complications, can)happen. For me, I went to the hospital and thought the baby will die. The doctor put me on a drip so that I could get better. They acted fast and I was written a letter and transferred to a bigger hospital." – Woman, Lundazi District | (m) "They [health facility staff] will help you when a baby is in breach position. They will find ways of delivering a normal baby. But if you deliver from home and it [a breach delivery] happens, by the time you go to look for transport, you would have been dead." – Woman, Mansa District |
| MWH availability | Barrier | No themes emerged | No themes emerged |
| | Facilitator | (n) "We mostly use ox-carts, but it is always better to go early and stay at the MWH until the day you deliver. I personally decided early that I needed to go to the MWH in case of difficulties with transport arrangements." – Woman, Kalomo District (o) "I considered the MWH. If I go there without pots I will be helped because they have pots there so I can cook before I deliver. Bed and blankets were also given." – Woman, Choma District | (p) "Usually that [MWH] is the first thing we look at. Due to long distances to the health facility in this area, women choose to go deliver at the clinics where there is a MWH, no matter the distance. Second, women know that if you are staying at the MWH, the health facility staff will be checking you regularly to see if you and the baby are ok and that you can deliver without any problems." – Woman, Lundazi District |

MHW, maternity waiting homes.

to buy all the delivery requirements including bleach, a bucket, gloves, chitenge (traditional wrapping cloth), baby blanket, baby clothes and nappies (table 3, quote g). Similarly, a woman in the intervention group and a woman in the control group discussed how health facility staff ignore or give less attention to patients who had not procured the required supplies. However, participants acknowledged that the health facility does not charge for the delivery itself (table 3, quote h) and there is no cost to stay at the MWH, which were frequently cited as a facilitator. Participants in both study arms perceived home delivery to cost more due to penalty fees, discussed further below (table 3, quote i).

### Penalties for home delivery

Penalties for home delivery arose as a factor that affects decision making for delivery location. Many participants discussed fearing home delivery, because they would have to pay either or both the health facility and to the village chief (table 3, quotes j and k). This factor emerged in both study arms and at all sites, though it was more salient for women in the control arm. Women reported payments to the chief in the form of a goat or chicken, and making monetary payments to the health facility staff (reported amounts between US$5 and US$25, depending on the study site). Women believed the 'law' came from either the government, the chief or the health facility, and is enforced by the chief, health facility staff and/or community health workers. When asked why the penalty exists, participants discussed preventing transmission of diseases such as HIV to the baby during delivery and preventing maternal mortality. A few participants also reported having to pay for their child's growth monitoring card at the health facility when they went for postnatal care after delivering at home and considered this a 'penalty' for their home delivery imposed by health facility staff. Attempts to avoid these penalties influenced women to delivery at health facilities.

Conversely, a few participants in the intervention arm believed the penalty fees for home delivery to be 'a hoax' (a story that nurses pass around), because they had never seen it enforced in their communities. This belief appeared to diminish the effect of the penalties on increasing facility deliveries.

### Safe delivery

No barriers related to safe delivery arose in either the intervention or control groups. Despite the perceived barriers of distance, transportation and cost, participants in both study arms regarded the health and well-being of the mother and baby as one of the top priorities for deciding where to deliver. Participants in both study arms emphasised the importance of delivering at a health facility as a facilitator, where trained health staff can help if any complications arose and expressed fear of dying from giving birth in their village (table 3, quotes l and m). Many women gave examples of complications such as the positioning of the baby (breach) or losing blood, and

they anticipated that the health facility had or would take care of them, give them blood or put them on 'a drip'. Some participants added that if the health facility staff were not able to handle the complications, they would be able to refer women to a higher-level facility.

### MWH availability

No barriers related to MWH availability arose in either the intervention or control groups. Participants in both study arms acknowledged that people in their communities consider the availability of an MWH when deciding where to deliver, especially if they live very remotely (table 3, quote n). This was more frequently discussed as a facilitator by participants in the intervention arm, where women also noted the importance of the amenities provided at the MWH, such as beds, blankets and cooking utensils (table 3, quote o). Although some participants in the control arm noted the availability of an MWH in their area, many wished for one of higher quality. Some participants even expressed their willingness to travel farther to a health facility with an MWH (table 3, quote p). However, some participants in the control arm reported they do not consider an MWH when deciding where to deliver; rather they are willing to travel to wherever they can deliver safely regardless of whether an MWH is present.

## DISCUSSION

This qualitative study explored differences in the factors remote-living women perceive as influencing the location of delivery in communities randomly assigned to receive the Core MWH model intervention and in communities randomly assigned to standard of care. We found that the availability and characteristics of the Core MWH Model did seem important to pregnant women: the MWH was more frequently mentioned as a facilitator by participants in the intervention arm, and women emphasised the amenities provided at the Core MWH Model, such as beds, blankets and cooking supplies.

At the same time, we found that some facilitators influencing delivery location were similar across both study arms. These included the perception of having access to a safer delivery, no user fees and avoiding penalties for home delivery. These findings are supported by other literature emerging from the same areas in Zambia.[19 23 24 43]

Despite implementation of the Core MWH Model and other demand-generating interventions including SMGL, women in both arms still perceived barriers to facility-based delivery, including distance and transportation, and costs associated with delivery, consistent with baseline evaluation findings.[19] Health facilities in these rural districts serve catchment areas of 5000–11 000 individuals.[44] Because of the geography and low population density in the rural areas, health facilities are spread out and those fully equipped to manage obstetric complications are even fewer. In the near future, human resource and cost requirements make it unlikely that the GRZ could feasibly bring health services, particularly basic emergency obstetric services, within 5 km of all

populations (a WHO core health service access indicator for health systems).[45] Government officials in Zambia and other countries with similar access issues should consider MWHs an alternative option to make health facilities physically accessible to the most rural populations. Though MWHs address the distance barrier, they alone do not address the transportation barriers that persist here and that have been widely reported in areas similar to rural Zambia.[4 46 47] Policy-makers should apply systems thinking to adopt financing policies that will provide benefits to rural residents, and expand monitoring for urban/rural inequities.[48 49]

Not surprisingly, MWH availability was more heavily discussed among participants in the intervention arm where the Core MWH Model was implemented, though the theme also emerged among control participants, indicating an overall awareness of MWH in the study areas. MWHs help women to overcome the distance and transportation barriers, allowing them to better plan and prepare to be close to health facilities when nearing delivery. Additionally, the proximity of MWHs to health facilities theoretically allows women to be easily attended by staff during the final weeks of pregnancy and when labour begins, increasing the likelihood for a safe delivery, one of the three perceived major drivers affecting women's decision to deliver at health facilities. A quantitative study in Zambia showed that health facilities with any MWH or waiting space, regardless of quality, had a slightly higher proportion of deliveries compared with those that had no waiting space, however, women were even more likely to deliver at health facilities with higher-quality MWHs versus those with low-quality MWHs.[14]

Women also explained that penalties for home deliveries influenced their decision making, saying they would deliver at a health facility to avoid paying penalties to the health facility or to traditional leadership, consistent with findings from two previous studies in rural Zambia.[31 50] However, some women noted they did not witness any penalties being enforced in their communities. Similarly, when rural Zambian women were previously asked to enumerate all costs associated with delivery, penalties among women delivering at home were insignificant compared with all other delivery costs.[51] Penalties may exist only as a rumour within these populations, or are levied in limited geographic areas. Further informal queries revealed that levying penalties for home delivery is not a law or official GRZ regulation and that the government discourages such practices. Regardless, the belief that penalties for home delivery exist is a motivator for women to deliver at a facility.

The concern over the cost of facility-based delivery also persisted. Zambian women are required by facility staff to provide delivery supplies, and they perceive the need to provide new items, particularly baby clothes, when delivering at a health facility to avoid embarrassment or being shouted at by staff.[43] A study in rural Zambia also found that women who had saved more before coming to a health facility for delivery were less likely to report feeling disrespected.[52] Previous research in these areas of rural Zambia found delivery supplies, such as clamps and cotton wool, represent a small percentage of the overall delivery expenditure, while baby clothes and a baby blanket were the drivers of expenses.[51] Consistent with our qualitative findings, this quantitative study also showed that expenditure differed by delivery location, with women delivering at facilities reporting higher expenditures than those delivering at home.[51]

Women perceived MWHs may reduce access barriers due to distance, facilitating increased use of delivery and postnatal health services among remote-living women corroborating the quantitative findings from this study.[26] However, perceived challenges with transportation and costs associated with delivering at a facility or using an MWH pose a potential barrier to their use, consistent with other studies.[43 53] Interventions that effectively address these challenges should continue to be explored and implemented.

### Study strengths
This study has several strengths. First, it provides insight on how MWH characteristics (ie, Core MWH Model vs existing MWHs) may influence perceptions about barriers to facility-based delivery. Second, because we did not know the range of variation to expect in our population, our method of using a randomly selected, representative sample of the population assured the best opportunity for diversity in perceptions.[54 55] Lastly, the rural population sampled for this study is a hard-to-reach population that likely faces the greatest barriers to facility delivery due to their low socioeconomic status, poor road infrastructure, limited transportation options, and low densities of facilities and skilled birth attendants. Using the qualitative data collected through this strategy, we were able to more deeply understand what influences these women's delivery location across seven culturally diverse and geographically dispersed study districts, providing important information for policy makers and programme implementers interested in serving these vulnerable populations.

### Study limitations
While this study has multiple strengths, there were also several limitations. First, our findings are subject to recall bias as we interviewed women who delivered up to 12 months prior to data collection. However, we primarily asked about what influences the location of delivery for 'women in this community,' which asks about social norm and is less subject to recall bias than personal experiences and decision making.[54] Second, it is likely that the presence of this study resulted in increased awareness of MWHs among control populations, as shown by the main quantitative study findings published separately.[22] We conducted multiple rounds of data collection for both the impact (baseline and endline) and implementation studies where community members and health system staff were asked about MWHs. Third, we did not disaggregate by MWH users and non-users or by location of MWH stay or delivery location. All these personal experiences likely influenced women's responses. We sought to understand

how community-level perceptions changed over time, not just individual-level decision making. A different analysis could be conducted to understand responses from women based on their MWH use or delivery location. Lastly, our intervention was implemented in a specific policy environment, and in SMGL-supported districts, limiting generalisability beyond the study districts.

## CONCLUSION

High-quality MWHs can help to address some of the well-known perceived barriers to facility-based delivery; however, other obstacles persist. While MWHs may be useful to address the distance barriers, no single intervention is likely to address all the barriers experienced by remote, rural and poor populations. The quantitative and qualitative results from this study suggest MWHs could be considered in a broader systems approach to improving access in remote areas by policy holders and implementers.

**Author affiliations**
¹Global Health, Boston University School of Public Health, Boston, Massachusetts, USA
²Research, Right to Care Zambia, Lusaka, Zambia
³University of San Francisco - School of Nursing and Health Professions, San Francisco, California, USA
⁴mothers2mothers, Lusaka, Zambia
⁵Programmes, Amref Health Africa, Lusaka, Zambia
⁶Office for Global Affairs & PAHO/WHO Collaborating Center, University of Michigan School of Nursing, Ann Arbor, Michigan, USA
⁷Africare Zambia, Lusaka, Zambia
⁸Health Behavior & Biological Sciences, University of Michigan School of Nursing, Ann Arbor, Michigan, USA
⁹Section of Infectious Diseases, Department of Medicine, Boston University School of Medicine, Boston, Massachusetts, USA
¹⁰Project Hope, Bethesda, Maryland, USA
¹¹Pediatric Centre of Excellence, National Health Research Authority, Lusaka, Zambia

**Acknowledgements** We appreciate the efforts, commitment and contributions of the project staff, administrative support staff, the data collection team and interviewers. From the Boston University/Right to Care team, we are thankful for the contributions of Denson Chongwe, David Kalaba, Deophine Bwalya, Parker Chastain, Allison Juntunen, Kathleen McGlasson, Elizabeth Ragan, Carey Howard, and Jason Park. From the Africare/University of Michigan team, we appreciate the efforts of Anne Naggyi, Isaac Sakala, Jessy Mtenje, Elizabeth Simwawa, Tenford Phiri, Lupiya Chilambwe, Jameson Kaunda, Nchimunya Chiboola and Nancy Lockhart. We are thankful to the team of translators and transcribers, and the two qualitative data coders Ingrid Olson and Jordan Musante for their hard work. We acknowledge the ongoing support of the Zambian Ministry of Health, the Provincial Health Offices, the District Health Offices; we thank Mary Nambao for her support and advice. We thank the Chiefs who welcomed us into their communities. This study could not have happened without the tireless efforts of the health facility staff and community-based volunteers who were champions of the intervention, advisors to project staff, and community guides to data collection teams. Finally, we thank the women, their infants, and their families for welcoming data collection teams into their homes and participating in this study. This research was conducted with funding from MSD for Mothers, the Bill & Melinda Gates Foundation, and The ELMA Foundation.

**Contributors** Co-principal investigators NAS and JRL acquired funding, conceived the research question, and designed the trial. Along with NAS and JRL, co-investigators PCR, DHH, TV, MLMK, EAM, and GB contributed to conceptualization of the study, development of the research questions, and design of the trial, sampling methodology, and instruments. NAS, JK, RMF, TV and TN oversaw investigation, research methodology and data acquisition. JK, RMF, TN, MB and VRS were responsible for data curation. RMF conducted the data analysis. TN, GB and GM participated in study coordination provincially and nationally. MB, VRS and GM implemented the interventions and oversaw project administration. RMF, JK, KJK, TV and NAS wrote the original draft of this manuscript. NAS is guarantor. All authors reviewed and edited the manuscript and approved the final version.

**Funding** This program was developed and implemented in collaboration with Merck for Mothers, Merck's 10 years, US$500 million initiative to help create a world where no woman dies giving life. Merck for Mothers is known as MSD for Mothers outside the United States and Canada (MRK 1846-06500.COL). The development of this article was also supported in part by the Bill & Melinda Gates Foundation (OPP1130334) https://www.gatesfoundation.org/How-We-Work/Quick-Links/Grants-Database/Grants/2015/06/OPP1130334 and The ELMA Foundation (ELMA-15-F0010) http://www.elmaphilanthropies.org/the-elma-foundation/. Funding for Online Open publication supported by Chronos Support through the Bill & Melinda Gates Foundation.

**Disclaimer** The findings and conclusions are those of the authors and do not necessarily reflect the policies or positions of the funders.

**Competing interests** None declared.

**Patient and public involvement** Patients and/or the public were not involved in the design, or conduct, or reporting, or dissemination plans of this research.

**Patient consent for publication** Not applicable.

**Ethics approval** This study involves human participants and was approved by Boston University Medical Campus Institutional Review Board Protocol ID: H-34526, ERES Converge IRB in Zambia, Protocol ID: 2015-Dec-012. Participants gave written informed consent to participate in the study before taking part.

**Provenance and peer review** Not commissioned; externally peer reviewed.

**Data availability statement** Data are available in a public, open access repository. https://hdl.handle.net/2144/44014

**ORCID iDs**
Rachel M Fong http://orcid.org/0000-0003-0296-2521
Jeanette L Kaiser http://orcid.org/0000-0001-6008-5219
Thandiwe Ngoma http://orcid.org/0000-0003-2643-5021
Taryn Vian http://orcid.org/0000-0002-6968-7002
Misheck Bwalya http://orcid.org/0000-0002-9879-7067
Viviane Rutagwera Sakanga http://orcid.org/0000-0002-5197-0387
Jody R Lori http://orcid.org/0000-0003-0564-5783
Kayla J Kuhfeldt http://orcid.org/0000-0001-9284-5307
Gertrude Musonda http://orcid.org/0000-0001-8477-0675
Michelle Munro-Kramer http://orcid.org/0000-0002-4298-6790
Peter C Rockers http://orcid.org/0000-0002-2198-127X
Davidson H Hamer http://orcid.org/0000-0002-4700-1495
Eden Ahmed Mdluli http://orcid.org/0000-0001-8973-7663
Godfrey Biemba http://orcid.org/0000-0002-6064-6722
Nancy A Scott http://orcid.org/0000-0002-4713-4642

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
