## [Reviewer comments · BMJ Open]

ARTICLE DETAILS

TITLE (PROVISIONAL)	Barriers and facilitators to facility-based delivery in rural Zambia: A qualitative study of women’s perceptions after implementation of an improved Maternity Waiting Homes intervention
AUTHORS	Fong, Rachel; Kaiser, Jeanette; Ngoma, Thandiwe; Vian, Taryn; Bwalya, Misheck; Sakanga, Viviane; Lori, Jody; Kuhfeldt, Kayla; Musonda, Gertrude; Munro-Kramer, Michelle; Rockers, Peter; Hamer, D.; Ahmed Mdluli, Eden; Biemba, Godfrey; Scott, Nancy

VERSION 1 – REVIEW

REVIEWER	Dirk Essink Amsterdam Publ Hlth Inst
REVIEW RETURNED	07-Dec-2021

GENERAL COMMENTS	An interesting study, worthy of publishing, with minor revisions. I would have been more interesting to also gain some information on the perceived quality/satisfaction with the MWHs. If this is part of another article than it would be prudent to mention this more specifically in the introduction / or integrate articles – as the (novel) contributions to the scientific field of access to care are limited in this paper – though still relevant. Introduction <line 39-46> reference is made to the two quasi-experimental studies that were done in relation to the MWH. However, no report is made of the actual findings of the studies. It would be prudent to share some of these results, as the impact of these MWH are a likely cause of changing perspectives. And if there are any insights into a (qualitative) process assessment of the homes this would also be extremely relevant. As the satisfaction with these MWH also influences perceived barriers – when such a process evaluation appears to be part of the current study make this explicit. The aim/question can be stated more explicit Methodology In the description of the intervention, please indicate if (no) costs were involved for the residing pregnant person. Or, in what other way the intervention was financed, and how they are sustained. Study design: please indicate the timeframe when households should have changed there perceptions regarding barriers. Only the time of data collection is presented (2018). And this is particularly relevant as ‘perceptions need to change – and exposure to this structural change is relevant’. This is even more relevant as ‘utilisation of MWHs” was not a selection criteria. No reference is made to the nature of questions asked/interview guideline. What was the basis of the interview guideline, the HBM? Or was it part of a larger evaluation? There is reference made to the highly dubious term In-Depth-Interviews (which are extremely vague
--

in what they mean), but were these open, semi-structured, how long did they take, and in line with the above, what topics did they cover. (please add the guideline in supplementary files)

The HBM model was used, but this is not visible from the results nor the discussion. In what way was it supportive and do you really need to explain it when it is of so little significance throughout the paper?

Results

<line 36> reference is made to the number of participants that used the MWHs. Apparently 40% of the control group also used these, where these similar? The same? Or don't you know. As it may very well affect your barriers. Similarly the high use of facility based delivery in both groups also makes analysis of barriers rather indirect – they were able to overcome-

<page 17 line 8> here you refer to cost for delivery, although they are actually cost associated with MWHs (the supplies). Of course this is part of the total cost for facility based delivery, but for analytical reasons I would definitely separate these.

Additionally, though associated with costs, you refer to acceptability of services when it concerns stigma/shame 'not being able to buy supplies' and 'poor attitudes' of staff towards those that are poor. I understand that you can't engage with all topics, I do feel that merging everything into few categories reduces the richness' of the data, and superficial reading may lead to missing some key issues. Penalties for home-delivery: the first sentence - No barriers related to penalties for home delivery arose in either the intervention or control groups.- is a bit odd. Although the penalty system itself is not well explained. It feels that it happens, or at least is feared.

Regardless the starting sentence and the text itself is not clearly explained. Penalties are given, which forces people for facility delivery, at the same time, some indicated that those penalties don't exist? But what has this to do with barriers to facility based delivery. Or is the barrier that it feels 'forced' and similarly to the current 'soft force on COVID-vaccination' leads to resentment. Or do you see the penalties for home delivery as a facilitator (better said driver) for facility based delivery.

Discussion:

The authors remark how the study supports earlier findings, but it is not clear about what is new. To my understanding and knowledge of the literature, the most interesting finding is that it appears that MWHs are supportive in increasing access. As they were utilised by approx. 60% of those in the interventions group and also a staggering <40% in the control group. Perhaps this is more important for increasing safe delivery than access. (perhaps quality is addressed in another article). I would strongly advise to be more clear about what is 'really relevant'

I miss a bit recommendations that are explicit to stakeholder groups.

Strengths & Limitations

In the strengths and limitations reference is made to three unique contributions. First I wouldn't call it contributions per se; arguably they are strengths – or the methods chosen allow for a strong validation. They are not contributions to the knowledge base (which remains rather similar) nor are they per se contributions to methods. They are done before. Although I expected and respect that you mention them as strengths, I think that the first assessment: data was collected during an intervention actually has two problems (1) was the time frame long enough to really change perspectives. And (2) if perspective changed was this due to the presence of the

	MWHs, or because the presence of the research team / efforts invested in the MWH Also, random selection does not yield much insight into whether the existence of the MWH reduces barriers. I really think that more purposive sampling of those that have utilised the MWH and those who haven't could have yielded more insights (this is actually still possible with stratifying the data into four groups: intervention vs control; MWH use vs non MWH use. Also, the control group also visited the MWHs quite often, so the randomness, and intervention areas were not that relevant apparently. Conclusions: I support the general finding that MWHs contribute to removing barriers, however, this was rather found in the generic perspectives and not in the comparison of the groups (hence, were using these two groups/randomisation really a strength? – or just a pragmatic choice as it ran next to other studies) Contributions: It is not my place to assess the contributions of each of the authors, but I doubt whether all 15 authors contributed in an 'academic fashion' to the paper. In my own institutional setting we are quite stringent on this.
--	---

REVIEWER	Indira Narayanan Georgetown University Medical Center, Pediatrics/Neonatology
REVIEW RETURNED	11-Dec-2021

GENERAL COMMENTS	 1. The article needs to indicate what were to the differences between these improved MWHs and the earlier ones 2. It is noted that consent taken for the surveys when necessary. Does this mean that it was not taken for all participants? Were the participants assured of confidentiality? 3. It may be difficult for some to understand the application of the word "facilitator" to a statement on penalties. It becomes clearer in the section on "Discussion". It may be better to explain this earlier under methodology. 4. May be worthwhile to rephrase the conclusion to make more meaningful, at least in the main narrative, if not in the abstract, and highlight what MTWs address and what they do not, in facilitating facility deliveries.
--

VERSION 1 – AUTHOR RESPONSE

Reviewer 1	
It would have been more interesting to also gain some information on the perceived quality/satisfaction with the MWHs. If this is part of another article than it would be prudent to mention this more specifically in the introduction / or integrate articles – as the (novel) contributions to the scientific field of access to care are limited in this paper – though still relevant.	There was a lot of rich content that came out of these interviews and we were unable to cover everything in this article due to the word limit. Perceived quality of the MWHs is being covered in a separate publication which is currently in progress. To highlight some of the findings: Women in the intervention group cited MWH quality (well-built structure, has needed amenities and facilities) as a top reason to utilize a MWH other than being close to the health facility before delivery and being attended to by

	health staff during their stay. Poor MWH quality was not cited as a reason not to use a MWH in the intervention group, whereas poor MWH quality remains a barrier amongst women in the control group. Standards of quality increased amongst women in the intervention group after the implementation of the Core MWH Model, whereas having a place to sleep (no bed or mattress) was satisfactory for women in the control group.
Introduction <line 39-46> Reference is made to the two quasi-experimental studies that were done in relation to the MWH. However, no report is made of the actual findings of the studies. It would be prudent to share some of these results, as the impact of these MWH are a likely cause of changing perspectives.	The key findings from the two quasi-experimental studies have been highlighted in the introduction section on page 6, lines 20-21.
Introduction: And if there are any insights into a (qualitative) process assessment of the homes this would also be extremely relevant. As the satisfaction with these MWH also influences perceived barriers – when such a process evaluation appears to be part of the current study make this explicit.	We do have findings from a mixed-methods sub-study of this larger process/implementation effectiveness evaluation looking at quality and utilization of MWHs in referral facilities providing comprehensive emergency obstetric and neonatal care. We conducted two focus group discussions with pregnant women staying at the improved MWHs (Core MWH Model – the intervention) at these referral facilities at two timepoints after the implementation of the intervention. Findings indicated that intervention MWHs had a significantly higher quality score than those at comparison sites after the implementation of the intervention. We also saw increases in MWH utilization at both intervention and comparison sites, but greater percentage increase at one of the two intervention MWHs. The process evaluation is made explicit in the introduction section, along with a reference and key findings from the above-mentioned sub-study looking at quality and MWH utilization at referral facilities. (pages 6, lines 21-24, page 7 lines 1-4)
Introduction: The aim/question can be stated more explicit.	We have adjusted the aim to be more explicit (page 7, lines 3-7). The aim of this article is to qualitatively explore remote-living women’s perceived barriers and facilitators to delivering at a health facility and compare the differences in perceptions between those who living in areas which received an improved Maternity Waiting Homes intervention and those living in areas that

	have existing maternity waiting homes which have not been improved.
Methodology: In the description of the intervention, please indicate if (no) costs were involved for the residing pregnant person. Or, in what other way the intervention was financed, and how they are sustained.	More details about the intervention, including financing and sustainability, were added to the intervention description section on page 8, lines 5-22.
Study design: please indicate the timeframe when households should have changed their perceptions regarding barriers. Only the time of data collection is presented (2018). And this is particularly relevant as 'perceptions need to change – and exposure to this structural change is relevant'. This is even more relevant as 'utilisation of MWHs" was not a selection criteria.	The intervention MWHs were launched shortly after the baseline study and operated for a minimum of 13 months before we conducted the endline study. Baseline occurred between March and May of 2016 and endline occurred between September and October of 2018. This timeframe has been added to the study design section on page 9, lines 4 and 15-17.
Methodology: No reference is made to the nature of questions asked/interview guideline. What was the basis of the interview guideline, the HBM? Or was it part of a larger evaluation? There is reference made to the highly dubious term In-Depth-Interviews (which are extremely vague in what they mean), but were these open, semi-structured, how long did they take, and in line with the above, what topics did they cover. (please add the guideline in supplementary files)	More details on the basis of and the types of questions asked in the semi-structured in-depth interview guide was added under the study design section on page 9, lines 9-15. The interviews took approximately 30 minutes. This is stated under the data collection section on page 10, line 5. The semi-structured in-depth interview guide has been added as a supplementary file.
Methodology: The HBM model was used, but this is not visible from the results nor the discussion. In what way was it supportive and do you really need to explain it when it is of so little significance throughout the paper?	We agree with your point that although we used the HBM initially, the results and discussion did not center on the HBM constructs. We have edited the methods section (page 10, lines 21-24) accordingly.
Results <line 36> Reference is made to the number of participants that used the MWHs. Apparently 40% of the control group also used these, where these similar? The same? Or don't you know. As it may very well affect your barriers.	We have added more details about the condition of the control group MWHs on page 9, line 2. MWHs at the control sites were of poor quality (e.g. leaky roof, no beds or mattresses, no lights, no cooking supplies) compared to the Core MWH Model that we had implemented at intervention sites. Some control sites did not have a formal space (e.g. women waited outside or the health facility let women wait in the wards). The improved MWH at the intervention sites (the Core MWH Model) was designed based on formative research and community input, and aimed to reduce common barriers to MWH utilization, such as poor infrastructure,

	lack of amenities, safety concerns and cultural issues. The hypothesis was that by improving these common problems with MWHs, it would further increase MWH utilization, which in turn increases facility delivery. The results do show that despite implementation of the Core MWH Model and other existing interventions aimed at increasing facility delivery, perceived barriers to facility delivery still exist. During the course of the study, there was a general increase in awareness of MWHs which may explain why we are seeing over 40% of women in the control group who have utilized a MWH, albeit of low quality. Please see page 20, lines 18-22, where we highlighted the findings of another study that supports this explanation.
Results: Similarly the high use of facility based delivery in both groups also makes analysis of barriers rather indirect – they were able to overcome.	As mentioned in the study setting (page 7, line 11), the study was conducted in areas that also received the Saving Mothers Giving Life initiative, which aimed to increase facility delivery rates. This initiative is described in the introduction (page 5, lines 21-24 and page 6, lines 1-4). We chose these areas on purpose as we wanted to ensure the health facilities could meet demand the MWHs may create. Results from the full household survey sample showed an unexpectedly high baseline facility delivery rate of 81% in the intervention group and 82% in the control group, and a facility delivery rate of 91% in the intervention group and 88% in the control group. The increase in facility delivery in the intervention group was statistically significant. As said in the above response, despite implementation of the Core MWH Model and other existing interventions, pregnant and recently delivered women in these remote, rural communities still perceive the common barriers to facility delivery.
Results <page 17 line 8> here you refer to cost for delivery, although they are actually cost associated with MWHs (the supplies). Of course this is part of the total cost for facility based delivery, but for analytical reasons I would definitely separate these. Additionally, though associated with costs, you refer to acceptability of services when it concerns stigma/shame 'not being able to buy supplies' and	We have added costs associated with MWH stay, and stigma related to costs, as sub-themes under the main theme of costs associated with delivery in Table 2 as well as in the narrative on page 17, lines 2-4 to highlight the nuances to the theme but also show that they are relevant together. When asked about costs associated with delivery during the interview, our sample

'poor attitudes' of staff towards those that are poor. I understand that you can't engage with all topics, I do feel that merging everything into few categories reduces the richness' of the data, and superficial reading may lead to missing some key issues.	population generally treated costs associated with delivery and costs associated with their MWH stay interchangeably. When we separated these for analytical reasons, the majority of the responses under this theme were about cost of transportation and delivery supplies, and not as many responses were directly about costs associated with staying at a MWH. The respondents did not discuss stigma generally (it did not arise as a theme on its own), but always in association with the ability to purchase delivery supplies for facility delivery. We thought it would be important to provide more depth to the results on costs associated with delivery, hence why we put these two topics together. It is a nuance of costs associated with facility delivery.
Results: Penalties for home-delivery: the first sentence - No barriers related to penalties for home delivery arose in either the intervention or control groups. Is a bit odd. Although the penalty system itself is not well explained. It feels that it happens, or at least is feared. Regardless the starting sentence and the text itself is not clearly explained. Penalties are given, which forces people for facility delivery, at the same time, some indicated that those penalties don't exist? But what has this to do with barriers to facility based delivery. Or is the barrier that it feels 'forced' and similarly to the current 'sot force on COVID-vaccination" leads to resentment. Or do you see the penalties for home delivery as a facilitator (better said driver) for facility based delivery.	We further explain the penalties in the discussion section on page 21, line 5-7. We recognize that the use of barriers in regards to penalties for home-delivery is confusing. We have reworded it as a factor that influences the choice for delivery location (page 17, line 16). Fear of penalties, whether or not those penalties were actually enforced, was a driving factor for choosing facility delivery, because women feared getting charged for home delivery. The penalties are not an official law of the Zambian government, but they may be levied locally some rural areas.
Discussion: The authors remark how the study supports earlier findings, but it is not clear about what is new. To my understanding and knowledge of the literature, the most interesting finding is that it appears that MWHs are supportive in increasing access. As they were utilised by approx. 60% of those in the interventions group and also a staggering <40% in the control group. Perhaps this is more important for increasing safe delivery than access. (perhaps quality is addressed in another article). I would strongly advise to be more clear about what is 'really relevant'	We have edited the Discussion section to make the relevance of our findings more clear (page 19, lines 12-20).
Discussion: I miss a bit recommendations that are	We specified the recommendation that

explicit to stakeholder groups.	MWHs could be considered in a broader systems approach to improving access to facility delivery/safe delivery in remote areas is for policy holders and implementers in the Discussion section (page 20, lines 5-6 and lines 9-10) and in the conclusion (page 23, line 10).
Strengths & Limitations: In the strengths and limitations reference is made to three unique contributions. First I wouldn't call it contributions per se; arguably they are strengths – or the methods chosen allow for a strong validation. They are not contributions to the knowledge base (which remains rather similar) nor are they perse contributions to methods. They are done before. Although I expected and respect that you mention them as strengths, I think that the first assessment: data was collected during an interventions actually has two problems (1) was the time frame long enough to really change perspectives. And (2) if perspective changed was this due to the presence of the MWHs, or because the presence of the research team / efforts invested in the MWH?	We have changed the language of “unique contributions” to strengths (page 22, line 2). Regarding the point about how to interpret timing of intervention, we have edited the limitations (page 22, lines 18-22).
Strengths and limitations: Also, random selection does not yield much insight into whether the existence of the MWH reduces barriers. I really think that more purposive sampling of those that have utilised the MWH and those who haven't could have yielded more insights (this is actually still possible with stratifying the data into four groups: intervention vs control; MWH use vs non MWH use. Also, the control group also visited the MWHs quite often, so the randomness, and intervention areas were not that relevant apparently.	We have revised the text on page 22, lines 2-6 to make the strength clearer, and to site references. We believe that it was important to sample a wide range of respondents who could provide rich information on barriers and facilitators to facility-based delivery, including those related to MWH use in intervention and control areas. When the range of variation in perception is not known, it can be advisable to use random sampling.
Conclusion: I support the general finding that MWHs contribute to removing barriers, however, this was rather found in the generic perspectives and not in the comparison of the groups (hence, were using these two groups/randomisation really a strength? – or just a pragmatic choice as it ran next to other studies)	See prior edits in Discussion section (page 19, lines 10-20) for how we have tried to make clear what is the contribution of the comparison between the two groups.
Contributions: It is not my place to assess the contributions of each of the authors, but I doubt whether all 15 authors contributed in an 'academic fashion' to the paper. In my own institutional setting we are quite stringent on this.	This was a very large study with 40 study sites across 4 provinces in Zambia, and which included two implementing partners and two academic partners. We limited authorship to those that were most critically involved and contributed significantly to the implementation of the study and review of the

	manuscript.
Reviewer 2	
The article needs to indicate what were to the differences between these improved MWHs and the earlier ones.	More details about the intervention maternity waiting homes (The Core MWH Model) was added to the intervention description section on page 8, lines 4-22.
It is noted that consent taken for the surveys when necessary. Does this mean that it was not taken for all participants? Were the participants assured of confidentiality?	Thank you for pointing out the need for clarification on the informed consents and assents under the ethical considerations section on page 11, lines 22-24. We clarified that informed consent was obtained from all participants, not only when necessary. Only assent was obtained when needed, for women 15 years old or younger.
It may be difficult for some to understand the application of the word “facilitator” to a statement on penalties. It becomes clearer in the section on “Discussion”. It may be better to explain this earlier under methodology.	We recognize that the use of facilitator and barrier in regards to penalties for home-delivery is confusing. We have reworded it as a factor that influences the choice for delivery location (page 17, line 16 and 18). Fear of penalties, whether or not they were actually enforced, was a driving factor for choosing facility delivery, because women feared getting charged for home delivery.
May be worthwhile to rephrase the conclusion to make more meaningful, at least in the main narrative, if not in the abstract, and highlight what MWHs address and what they do not, in facilitating facility deliveries.	We have edited the Discussion and the Conclusion based on similar comments from Reviewer #1.